Evidence of prostate cancer-linked virus zoonoses from biophysical genomic variations

Alsufyani Daniah 1 2 sufyanid@ksau-hs.edu.sa
http://orcid.org/0000-0002-7310-8755 Lindesay James 3
1 College of Sciences and Health Professions, King Saud Bin Abdulaziz University for Health Sciences , Jeddah , Saudi Arabia
2 King Abdullah International Medical Research Center , Jeddah , Saudi Arabia
3 Howard University , Washington, D.C. , United States
Uversky Vladimir
Electronic publication date: 2024 Dec 6
Publication date: 2024
Volume: 12
Electronic Location ID: e18583
Received 2024 Jun 19; Accepted 2024 Nov 4
Copyright: © 2024 Alsufyani and Lindesay
Copyright year: 2024
Copyright holder: Alsufyani and Lindesay
License: This is an open access article distributed under the terms of the Creative Commons Attribution License, which permits unrestricted use, distribution, reproduction and adaptation in any medium and for any purpose provided that it is properly attributed. For attribution, the original author(s), title, publication source (PeerJ) and either DOI or URL of the article must be cited.
License URL: https://creativecommons.org/licenses/by/4.0/

Keywords: Adaptive force, Population health, SNPs, Genodynamics, Zoonotic viruses, Prostate cancer

Funding: The authors received no funding for this work.

==============================
An ongoing double-blind examination of (mathematically) smooth functional dependences of population-based genomic distributions of single nucleotide polymorphisms (SNPs) on quantified environmental parameters has flagged a SNP that has been associated with prostate cancer for dependence on zoonotic viruses. The SNP rs13091518 is an intergenic variant near the gene/pseudo-gene COX6CP6 on chromosome 3. The risk T allele, which is the major allele in all homeostatic populations considered, clearly demonstrates a negative adaptive force of about −0.1 universal genomic energy units/zoonotic virus unit. This biophysical perspective has thus provided evidence for a causative relationship between zoonotic viruses and prostate cancer. Our findings are consistent with other studies that have found an association between several zoonotic viruses and prostate cancer. This result demonstrates the significance of an intergenic variant in the adaptive response to a viral zoonotic pathogen.

Introduction

Since the inception of the HapMap project of population-based human genomes, access to the study of the subtleties and dependencies of the distribution of genomic variants has resulted in considerable insights into the dynamics of the factors that influence genomic variations. Since the data are specific to each population, this allows an examination of how quantifiable environmental influences can affect the populations subject to those influences. Due to migrations, human populations often smoothly adapt to environmental changes along their various migration routes. Some models have developed statistical free energy functions to describe the evolutionary dynamics of finite populations (Sella & Hirsh, 2005), where perturbations of a population’s free energy evolve toward “fitness peaks” for the variants (Agozzino et al., 2020). However, many such tools utilized in bioinformatics lack a universal genomic dimensional unit that can quantify the pressures on the genomic variants of a given population. In particular, genome-wide association studies (GWAS) only search for associations of single nucleotide polymorphisms (SNPs) with a particular disease or trait.

In our approach, genodynamics (Lindesay et al., 2013) develops a universal dimensional biophysical unit (a genomic energy unit) associated with genomic variations. The existence of this unit then allows adaptive forces (i.e., differences in genomic variant distributions between homeostatic populations due to environmental differences) to be ascertained. Genodynamics searches for meaningful relationships between the distributions of population-based genomic variants and quantifiable environmental influences. It seeks to model the information dynamics of bi-allelic SNPs due to those influences. The approach presently being undertaken involves a double-blind examination of all SNP variants on chromosome 3. The establishment of smooth functional dependencies of stable population distributions upon any impinging influence implies that all adaptations optimize the overall health of each population. This approach is thus absent any focus on disease. In this article, the dynamics of a particular common genomic variant between homeostatic populations that presently reside in their ancestral geographic homelands as a function of zoonotic viruses will be presented. Population homeostasis implies that several generations of the population have maintained the same distribution of genomic variants, which is analogous to equilibrium in thermodynamics. Subsequent to flagging rs13091518 with richness of zoonotic viruses, a literature search summarized in the discussion section revealed several biological studies indicating a direct link of the risk allele to various viruses associated with prostate cancer.

Methods

The dispersion of the human species throughout a wide variety of stable environments has generated a large diversity in both phenotypes and genotypes. The resulting genetic stability often exhibits Hardy-Weinberg equilibrium within each population, where the frequencies of allelic variants are maintained throughout several generations or other arbitrary subgroups (Hardy, 1908). These observations motivated the development of a biophysical approach that formulates the dynamics of the information codified within the populations’ genomes, enabling the investigation of meaningful relationships between environmental stimuli and genomic SNPs. This approach is referred to as “genodynamics”. For the present study (on-going for 4 years), we have chosen 10 populations genotyped within their ancestral geographic regions to search for any dependencies of maintained genomic order on a set of 15 quantifiable ancestral environmental parameters. Obtaining quantified environmental parameters remains one of the most challenging aspects of this study.

Genodynamics modeling approach

Geodynamics utilizes a well-established measure of the degree of maintained order within a dynamic system defined as the information content (IC) (Shannon & Weaver, 1963; Bergstrom & Lachmann, 2004; Susskind & Lindesay, 2004). A more portable measure of maintained order is the normalized information content (NIC). The NIC can be expressed in terms of the entropy of the distribution s=−∑a⁡palog2pa, which itself is defined in terms of the probabilities (frequencies) of variants a, here labeled pa. For the variants in a population’s genome, the NICgenome is defined as

NICgenome=smax−sgenomesmax,

where sgenome is the entropy of the whole genome, and smax is the entropy of maximal variation. The entropy of the complete genome of a population sums the maintained entropies of each SNP variant on all chromosomes. The maximum possible entropy of a human genome assigns maximum variation for each of those variants. When utilizing bi-allelic SNPs as the genomic variants, the maximum entropy is given by smax = NSNPs, where NSNPs is the total number of (biallelic) SNPs on the genome. From the NIC, a degree of intrinsic disorder analogous to the temperature in thermodynamics can be derived by minimizing the overall genomic free energy (Alsufyani & Lindesay, 2022). This dimensional environmental potential is defined by TE=μˇNICgenome, where the unit μˇ is the maximum SNP potential of a non-linked bi-allelic SNP and is assigned a value of 1 (human) genomic energy unit (GEU). From this relation, it is clear that populations with a greater degree of disorder have a higher TE.

Bi-allelic SNPs that are not in linkage disequilibrium have additive genomic potentials related to allelic frequencies pa(S) as defined via the expression

μa(S)=μˇ−TE−TElog2pa(S).

If only a single allele is shared amongst the whole population ( i.e.,pa(S)=1,pa¯(S)=0), then SNP potential is fixed at the fixing potential μˇ−TE. As previously mentioned, these potentials take the value of 1 GEU for bi-allelic SNPs of maximum variation (Alsufyani & Lindesay, 2022). For a particular SNP (S), the SNP potential μ(S) is just the population averaged value of its allelic potentials μ(S)=∑a⁡pa(S)μa(S). For the populations examined in what follows, the environmental potentials TE are given in Table 1. The characterization of the environmental potential TE requires evaluation of the whole genome information content of each population under consideration. A previous exploration of the (somewhat smaller) whole genome information content of HapMap populations convinced the authors that the information contents of chromosome 3 accurately represents that of the whole genomes within a percent (Dunston et al., 2014).

Table 1 Environmental potentials of the examined populations.

Population	T E	
PEL	1.108	
CLM	1.125	
FIN	1.125	
KHV	1.106	
JPT	1.108	
TSI	1.122	
MSL	1.237	
CHB	1.108	
IBS	1.110	
LWK	1.225	

Allelic variants that are in linkage disequilibrium maintain a collective order unique to a given population that decreases the overall entropy of those variants. The collection of such SNPs form haploblocks with the various haplotypes defining the variants within the haploblocks. In the same manner that each allelic and SNP potential can be defined for an individual genomic locus, the haplotype potentials and haploblock potential μ(H) (which is the population average of the haplotype potentials) can directly be defined for a linked set of genomic loci (Alsufyani, 2024). One should note that the haploblock structure of any population is unique. However, most of the GWAS data that will be utilized to ascertain biological associations focus on individual SNPs. For this reason, we have developed a mechanism for meaningfully distributing the haploblock potentials amongst the SNPs (Lindesay et al., 2018): if a specific allele is fixed for all members of a population, its distributed SNP potential is defined to be the fixing potential μS(H)=μˇ−TE;

the sum of the distributed SNP potentials within a haploblock is the same as the haploblock potential ⟨μ(H)⟩=∑S⁡μS(H);

the overall haploblock potential is linearly distributed amongst its constituent SNPs proportionate with the occurrence of the alleles consistent with the previous requirements according to μS(H)=μfixed+[⟨μ(H)⟩−n(H)μfixed][P¯s′∑s′⁡P¯s′].

The increased degree of maintained order due to the linkage disequilibrium thus defines a ‘binding potential’ εbinding(S)=μS(H)−⟨μ(S)⟩ relative to the averaged SNP potential assigned if the SNPs were not linked. The distributed SNP potentials can finally be used to define distributed allelic potentials μas(H)=μas(S)+εbinding(S) which share the lowered genomic potential of the SNP.

We search for allelic potentials whose values vary smoothly between the populations in such a manner that simple mathematical forms assign those potentials as functions of quantified environmental parameters. Physical systems tend to slide down the slopes of potential energy, defining a force as the slope of the potential energy curve. In analogy, the adaptive force (or adaptive “pressure”) is defined as the slope down the gradient of an allelic potential curve as a function of the environmental parameter fa=−∂μa∂λ.

Genomic analysis in environmental context

The populations examined in this study were genotyped within their ancestral geographic environments. These include Peruvian in Lima, Peru (PEL), Colombian in Medellín, Colombia (CLM), Finnish in Finland (FIN), Kinh in Ho Chi Minh City, Vietnam (KHV), Japanese in Tokyo, Japan (JPT), Toscani in Italy (TSI), Mende in Sierra Leone (MSL), Han Chinese in Beijing (CHB), Iberian population in Spain (IBS), and Luhya in Webuye, Kenya (LWK). The populations were double-blind scanned for allelic dependencies on the 15 quantified environmental parameters including, altitude, humidity, pressure, temperature, rain, UVB, wind speed, and pathogens (virus, bacteria, helminth, protozoa) associated with zoonotic hosts (chiroptera, primates, rodentia, soricomorpha). Environmental data was population-averaged using a variety of cities distributed throughout the ancestral regions. Maps and non-tabular data were independently quantified by the authors, and averaged results are ultimately utilized in the searches. To this point, 95 thousand SNPs have been scanned, and only about 0.007% have flagged a meaningful dependency so far.

Criteria for flagging allelic potential dependencies

The formulation requires a smooth mathematical dependency on a specific environmental parameter for a result to be meaningful. For our purposes, only relatively simple dependencies of the allelic potentials on environmental parameters are being considered. During a given search procedure, the root-mean-squared (RMS) deviation of the distribution of the allelic potentials of the populations from optimized fitted curves is calculated. A smooth dependency of an allelic potential on an environmental parameter will only be flagged if the dimensionless ratio of the RMS deviation to the maximum change in that allelic potential is within 10%.

Results

Allelic potentials that quantify the pressures induced on genomic variants due to environmental influences continue to provide a useful tool for discovering the natural response to stimuli and pathogens that affect the given population. For the present study, only simple functional forms monotonic in allelic potentials (but not necessarily in the SNP potential) have been considered (Alsufyani & Lindesay, 2022, 2023). The SNP rs13091518 only flagged the populations for the pathogen ‘zoonotic virus richness’, where ‘richness’ is defined in the reference as follows (Alsufyani & Lindesay, 2023):

Richness: the number of unique species within a particular geographic area; richness is a count-based metric for quantifying diversity, which contrasts with other metrics, such as functional trait diversity (the different types of traits represented within a geographic area) or genetic diversity.

It is noteworthy that for rs13091518, the potentials of both alleles as well as the SNP itself flagged dependencies upon the zoonotic viruses. The type of flag of the SNP emphasizes a direct dependence of the SNP potential due to the pathogen. The risk allele T exhibits a negative adaptive force of about −0.1 GEUs/zoonotic virus unit due to the pathogen, consistent with an increasing risk to exposed populations.

Figure 1 demonstrates the SNP potential of rs13091518 vs. exposure to zoonotic viruses, which notably flagged with relative RMS deviation of less than 0.03. The SNP potential reflects the favorability of maintaining an overall degree of variation of both alleles within the population. In the figure, blue points represent SNPs that are not in linkage disequilibrium and red points represent SNPs in linkage disequilibrium. The figure clearly demonstrates that any presence of this pathogen breaks the healthy maintained correlated genomic variation of the SNP haploblock within the population PEL. The increased conservation of the block of SNPs is reflected in its considerably lower SNP potential when compared to the more exposed populations. There is some indication that the non-risk allele C begins to have an overall more favorable contribution to population health for the highly exposed populations due to the downturn in the potential curve.

Figure 1 The correlation between the rs13091518 SNP potential values (in GEUs) and the richness (number of species) of zoonotic virus pathogens.

Figure 2 illustrates the allelic potential of the T allele, which is the risk allele for prostate cancer. This allele remains the major allele in all the populations chosen for consideration. This risk is clearly demonstrated through the negative adaptive force of −0.1 GEUs/zoonotic virus unit, with a relative RMS deviation of 0.055, well within our flagging criterion. The T allele is highly conserved with an allelic potential of −0.37 GEUs for the population PEL, which has the lowest exposure to zoonotic viruses. This is in contrast to the allelic potential of nearly 1 GEU for the population TSI, indicative of highest allelic variation.

Figure 2 The correlation between T allelic potential (in GEUs) of rs13091518 and zoonotic viruses host richness. The adaptive force is about −0.1 GEUs/zoonotic virus unit.

Finally, Fig. 3 displays the allelic potential of the C allele, which (barely) remains the minor allele in all populations considered. It is clear from the plots that an increased presence of viral zoonoses induces higher variation in the SNP, which is provided by the C allele. This mitigation of risk translates into an adaptive force of +0.16 GEUs/zoonotic virus unit, with a relative RMS deviation of 0.06. The type of flag emphasizes a direct dependence of the frequencies of alleles in the population as opposed to the allelic potential itself. This indicates that the increased conservation of the C allele acts to ‘switch on’ an enhancement to population health.

Figure 3 The correlation between C allelic potential (in GEUs) of rs13091518 and zoonotic virus richness. The adaptive force is about +0.16 GEUs/zoonotic virus unit.

Discussion

The SNP rs13091518 is an intergenic variant on chromosome 3 at location 70747545(Conti et al., 2021). Genome wide association studies have relatively recently (2021) associated the T allele of this variant (the major allele in all of the examined populations) with an increased risk of prostate cancer (Sollis et al., 2023). The nearest gene/pseudogene, cytochrome c oxidase subunit 6C pseudogene 6 (COX6CP6), is located between 70751058 and 70751523. Some pseudogenes have been suggested to play a role in regulating protein-coding transcripts (Chan & Chang, 2014). More recent developments in sequencing technology have discovered a large number of pseudogenes that can interact with DNA, RNA, and proteins that modulate target gene expression. This indicates their potentially strong involvement in the development and progression of certain diseases, especially cancer (Xu et al., 2020).

It is noteworthy in our results that the only population for which this SNP is in linkage disequilibrium is that population with the lowest viral load. This loss of coordinated genomic variance with increasing viral zoonoses amplifies the increased virally induced disorder amongst the populations. This is consistent with the observed negative adaptive force associated with the risk allele T. Evidently, this risk major allele maintains a beneficial biological function moderated through its viral risk association. Previous examinations of the MHC region on chromosome 6 (associated with adaptive immunity) have demonstrated that regulatory regions display a lower degree of maintained order as quantified by the NIC when compared to genic regions within a given population (Lindesay et al., 2014).

Other studies support our hypothesis, suggesting a link between increased risk of prostate cancer and exposure to zoonotic viruses. A study in 2007 about prostate cancer pathogenesis involves both heritable and environmental factors. Environmental factors have been implicated in studies involving immigrant Asians living in Western countries as compared to their counterparts living in Asia (De Marzo et al., 2007). In a review article (Maia & Hansen, 2017), the authors have argued that viruses have been shown to play a significant role in the etiology of prostate cancer, as well as in promoting a pro-inflammatory microenvironment. Such microenvironments have been shown to be immunosuppressive and to support cancer progression.

Numerous studies have established a correlation between specific zoonotic viruses and an increased risk of developing prostate cancer, which will be elaborated upon in the following sections. Both human and non-human polyomaviruses consist of several types of non-enveloped DNA that encode oncogenes associated with human disease. The BK virus, which is a polyomavirus, has been found as an etiological cofactor of early-stage prostate cancer progression (Das, Wojno & Imperiale, 2008). This virus has been suggested to infect the urinary tract, encoding tumor antigens that inactivate the tumor suppressors p53 and Rb1. However, this virus is not detected in those cells exhibiting more advanced cancer progression. Another study examining the JC (John Cunningham) polyomavirus (JCV) as a risk factor in prostate cancer detected the presence of the JCV Large T antigen (L Tag) in cases vs. controls to assess its risk (Gorish, 2022). The study found a significant relationship between JCV infection and the probability of developing prostate cancer, with the JCV LTag in the cases group higher than that in the controls with a p-value of 0.006. In addition, viral load was also significantly higher with a p-value of 0.002.

Polyomaviruses encode oncogenes and are associated with cancer (Scuda et al., 2013). They are known to cause infections in patients that have compromised immune responses. Recent investigations suggest that certain polyomaviruses from non-human primates circulate among humans (Kuchipudi et al., 2021). Furthermore, the possibility of zoonosis (Scuda et al., 2013) from bovine polyomaviruses due to close human encounters is suspected. Human JCV has recently been found to cause multifocal leukoencephalopathy (Dugan, Gasparovic & Atwood, 2008). In humans, sera reactivity against BKV and JCV has been found to be cross reactive with simian virus 40 (or old-world monkey virus) (Scuda et al., 2013; Kean et al., 2009). BKV has been found to induce tumors in newborn hamsters and mice (Roka, Prieto & Gershenwald, 2008). Furthermore, some of the evidence suggesting that BKV is oncogenic has resulted from studies involving tumorigenesis in rodents inoculated with the virus (Ambalathingal et al., 2017). However, due to the cell specificity of BKV and narrow range of hosts, the development of animal models to study this infection is difficult (Borriello et al., 2022).

Kaposi’s sarcoma-associated herpes virus (KSHV), formerly known as humangammaherpesvirus 8 (HHV-8), is one of several known rhadinoviruses. KSHV has been found to be associated with prostate cancer amongst those who carry the IFNL4 gene. IFNL4 is a polymorphic pseudogene on chromosome 19, sometimes encoding the interferon lambda 4 protein. The HHV-8 virus has been associated with both aggressive and non-aggressive prostate cancer amongst those carrying at least one G allele in IFNL4 (Jenkins et al., 2023). Herpesviruses have been shown to cause several diseases in both humans and animals (Woźniakowski & Samorek-Salamonowicz, 2015). Of the human herpesviruses, KSHV is the most closely related to zoonotic rhadinoviruses (Dollery, 2019; McGeoch, Rixon & Davison, 2006). It is also the only rhadinovirus known to infect humans (Chang et al., 1994). Several studies indicate that KSHV can infect most human cell types, and several animal cell types (Pica & Volpi, 2007; Schäfer, Blumenthal & Katz, 2015). The range of hosts of herpesviruses is narrow. However, due to specifics in the kind of encoded nucleic acid, it is possible that herpesviruses can be transmitted to other species. Even if the presence of a herpesvirus in its natural host is asymptomatic or only manifests mild symptoms, its transmission to another species might lead to severe disease or death (Woźniakowski & Samorek-Salamonowicz, 2015).

Finally, it should be noted that the Zika virus has been found to actually decrease the proliferation of a prostate cancer cell line (Delafiori et al., 2019). This indicates the possibility that environmental exposure to this virus might be associated with a positive adaptive response, reflected by the enhanced conservation of the C allele.

Supplemental Information

Supplemental Information 1 SNPs Potentials.

Supplemental Information 2 Environmental Parameters - SNPs Potentials Relationships.

Additional Information and Declarations

Competing Interests

Author Contributions

Data Availability

The authors declare that they have no competing interests.

Daniah Alsufyani conceived and designed the experiments, analyzed the data, prepared figures and/or tables, authored or reviewed drafts of the article, and approved the final draft.

James Lindesay conceived and designed the experiments, analyzed the data, authored or reviewed drafts of the article, and approved the final draft.

The following information was supplied regarding data availability:

The codes for calculating SNPs and alleles potentials and to find the associations between environmental parameters and potentials are available in the Supplemental Files.

The SNP rs13091518 frequencies can be generated at the Allele Frequency Calculator: https://grch37.ensembl.org/Homo_sapiens/Tools/AlleleFrequency?db=core.

All the maps utilized for the virus richness data are available at Global Patterns of Zoonotic Disease in Mammals, https://doi.org/10.1016/j.pt.2016.04.007.

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
