# Peer review of "Evidence of prostate cancer-linked virus zoonoses from biophysical genomic variations"

_PeerJ, doi:10.7717/peerj.18583_

## Round 0.1 · original submission · Major Revisions

Please address issues pointed by the reviewers and amend manuscript accordingly

Reviewer 1 ·

Basic reporting

The manuscript presents a study that investigates the potential link between genomic variations in prostate cancer and exposure to zoonotic viruses. The authors introduce a novel biophysical metric, "genomic energy units" (GEUs), to quantify the influence of environmental factors, particularly zoonotic virus richness, on specific single nucleotide polymorphisms (SNPs). The study focuses on the SNP rs13091518 and examines its potential adaptive significance across different populations. The research explores how these biophysical genomic variations might contribute to the susceptibility to prostate cancer, offering a unique perspective on the interaction between environmental factors and genetic predisposition.
The referee would point out that there are some strengths on the manuscript. Firstly, the introduction of a biophysical metric to quantify the impact of environmental factors on genomic variations is innovative and offers a new dimension to understanding the genetic predisposition to diseases like prostate cancer. In addition, the study effectively integrates concepts from genomics, biophysics, and evolutionary biology, which could provide valuable insights into the complex interplay between genetics and environment in cancer etiology. Finally, by focusing on zoonotic viruses, the manuscript addresses a timely and relevant topic, especially considering the global attention on zoonotic diseases and their potential impact on human health.
The following are the weak points worth further addressing and investigation to improve the manuscript.
1. The manuscript is dense with technical jargon and complex concepts, which may limit its accessibility to a broader audience. Some of the biophysical concepts introduced may require further simplification or explanation to be understood by readers from diverse scientific backgrounds better understand the innovative aspects of the study.
2. The study presents a novel hypothesis but lacks experimental validation or empirical evidence to support the proposed link between SNP variations and zoonotic virus exposure. This significantly limits the conclusions that can be drawn from the study. Moreover, the study heavily focuses on a single SNP, rs13091518, which may limit the generalizability of the findings. While identifying the non-protein-coding gene COX6CP6 near the SNP, the authors may include some excellent examples to discuss the work, and could refer to but not limited to noncoding RNA relevant prostate cancer risk SNPs and relevant regulatory mechanisms underlying the prostate cancer post-GWAS study
3. In a similar vein, the manuscript could benefit from a more detailed discussion of the biological mechanisms underlying the observed associations and incooperating relevant function study of prostate cancer risk SNPs with the references as described above.
4. On line 182, Sollis, et al., 2022 reference is for the catalog of all GWAS results. The authors should replace this one with the exact publication decribing the SNP.

Experimental design

Please refer to 1. Basic reporting

Validity of the findings

Please refer to 1. Basic reporting

Additional comments

Please refer to 1. Basic reporting

Reviewer 2 ·

Basic reporting

I completely do not comprehend the merit of this manuscript due to several major flaws, including poor writing, unstructured introduction, and strange methodology that lacks explanation.

The text is poorly written. Specifically, sentences are often too long with many strange words, which are very difficult for audience to follow. For example, in line 52-54,

“In this study, the dynamics of the common genomic variants of homeostatic populations that presently reside in their ancestral geographic homelands as a function of a set of quantified physical environmental parameters will be examined. ”

The Introduction section is poorly organized. This manuscript is about zoonoses, prostate cancer, and genomic variations. But the Introduction section does not mention necessary background about zoonoses, prostate cancer, or the relationships between them. Besides, Introduction should briefly cover the methodology and conclusion, which are absent.

Experimental design

The primary research goal is missing, and the methodology overall is very difficult to understand. For example, the authors defined “normalized information content” (NIC), which can be calculated based on “entropy of the whole genome” (s_genome) and “entropy of maximal variation” (s_max). However, the authors did not continue to describe what are s_genome or s_max. There are many more strange terminologies like these that did not come with an explanation, such as “SNP potential”, “environmental potential”, “genomic energy unit”, “maximum variation”, “adaptive forces”, maintained order”, etc. All these terminologies are probably based on a model/method (so-called “Genodynamics”) that the authors previously published (Lindesay et al. 2014 Journal of computational biology and bioinformatics research). This paper, however, is not well-known (cited by 4 in 10 years), so the authors cannot just assume that the audience should know about this method. Most importantly, the authors did not explain why they used these metrics (i.e., what can these metrics tell us).

Validity of the findings

The validity of the findings is extremely difficult to assess because a) the research goal is not well-defined; b) the methodology is poorly explained and justified; and c) there lacks connections between the methods and the findings.

---

## Round 0.2 · Minor Revisions

Although we did not receive comments from the reviewers, your manuscript was evaluated by one of our Section Editors, who indicated that this work is based on previous research, but there are typos and especially problems with the references, so it is difficult to understand what exactly was done and the significance of the results. Therefore, we ask you to consider revising the manuscript to make it easier to refer back to the relevant literature.

Also, in the discussion, although the first few paragraphs relate the identified SNP to potential virus interaction, the remainder of the paragraphs are more difficult to understand how they relate to the SNP (either directly or indirectly). Please make the corresponding message clearer.

Please also address a number of typos - including:

A number of places where spaces are needed
Line 46, thus absent
Line 64, Suskind and Lindesay
Line 120, what reference is [6] (others are given by author)

All of the references need work, most do not have titles - a full bibliography is needed.

---

## Round 0.3 · accepted · Accept

All issues indicated by the Section Editor were addressed, and I confirm that the revised manuscript is acceptable now.